# A Systematic Review on Digital Soil Mapping Approaches in Lowland Areas

Odunayo David Adeniyi [1,*], Hauwa Bature [2] and Michael Mearker [1,3,4,*]

1 Department of Earth and Environmental Sciences, University of Pavia, Via Ferrata 1, 27100 Pavia, Italy

2 Department for Sustainable Development and Ecological Transition, University of Eastern Piedmont, Via Duomo 6, 13100 Vercelli, Italy; 20050726@studenti.uniupo.it

3 Leibniz Centre for Agricultural Landscape Research, Working Group on Soil Erosion and Feedback, Eberswalder Str. 84, 15374 Müncheberg, Germany

4 Consiglio Nazionale delle Ricerche, Institute for Georesources and Geodynamics, Via Ferrata 1, 27100 Pavia, Italy

* Correspondence: odunayodavid.adeniyi01@universitadipavia.it (O.D.A.); michael.maerker@zalf.de (M.M.)

**Abstract:** Digital soil mapping (DSM) around the world is mostly conducted in areas with a certain relief characterized by significant heterogeneities in soil-forming factors. However, lowland areas (e.g., plains, low-relief areas), prevalently used for agricultural purposes, might also show a certain variability in soil characteristics. To assess the spatial distribution of soil properties and classes, accurate soil datasets are a prerequisite to facilitate the effective management of agricultural areas. This systematic review explores the DSM approaches in lowland areas by compiling and analysing published articles from 2008 to mid-2023. A total of 67 relevant articles were identified from Web of Science and Scopus. The study reveals a rising trend in publications, particularly in recent years, indicative of the growing recognition of DSM's pivotal role in comprehending soil properties in lowland ecosystems. Noteworthy knowledge gaps are identified, emphasizing the need for nuanced exploration of specific environmental variables influencing soil heterogeneity. This review underscores the dominance of agricultural cropland as a focus, reflecting the intricate relationship between soil attributes and agricultural productivity in lowlands. Vegetation-related covariates, relief-related factors, and statistical machine learning models, with random forest at the forefront, emerge prominently. The study concludes by outlining future research directions, highlighting the urgency of understanding the intricacies of lowland soil mapping for improved land management, heightened agricultural productivity, and effective environmental conservation strategies.

**Keywords:** geostatistical approach; lowland; low relief; machine learning; SCORPAN; soil mapping





## 1. Introduction

Soil, as the foundation of terrestrial ecosystems, plays a crucial role in supporting agriculture, biodiversity, and ecosystem services [1]. In the pursuit of sustainable land management and informed decision-making, accurate soil information in the form of soil maps is paramount. Traditionally, soil mapping involved labour-intensive field surveys and manual data collection methods, which often present limitations in terms of spatial coverage, resolution, and efficiency [2]. However, the digital revolution has transformed soil mapping practices, paving the way for innovative approaches that harness the power of technology, data science, and remote sensing [3].

Digital soil mapping (DSM) has revolutionized the field of soil science by combining traditional soil survey techniques with modern computing technologies [4,5]. DSM creates and populates spatial soil information systems using field and laboratory observational methods coupled with spatial and nonspatial soil inference systems [6]. It combines soil science, geographic information science, quantitative methods, and cartography within a framework that utilizes environmental data to predict soil classes and properties [7]. In

recent years, we observe a substantial increase in DSM activities driven by (i) the increasing demand for quantitative and spatial soil information, (ii) the development of statistical models and artificial intelligence combined with computer resources to compute and store these data, and (iii) enormous advances in easily obtainable environmental variable data for the rapid production of soil class and property maps [5,8].

McBratney et al. [7] formulated the general framework of DSM which was built on Jenny's model (S = CLORPT) of soil formation [9], where S is the soil, and the acronym CLORPT stands for climate, organisms, relief, parent material, and time. CLORPT factors are soil-forming factors; however, McBratney et al. [7] added the spatial position "n" to Jenny's formulation and proposed the SCORPAN model for soil mapping. This updated equation provides a spatial model to quantitatively express the relationship between a soil property or class and the environmental variables for a given spatial location. Based on the first law of geography and soil genesis theory, geostatistical and soil landscape models have been extensively explored in local, regional, and global DSM.

However, most DSM studies have focused on areas such as high-relief land [10–12], where terrain and vegetation exhibit certain spatial variations and correlate with soil spatial patterns. In this rapidly evolving sector of soil science, one specific terrain or landscape that demands careful consideration is lowland areas. Lowlands, encompassing floodplains, deltas, and coastal regions, are dynamic and complex landscapes shaped by complicated interactions between land, water, and ecosystems. Lowlands represent extensive, ecologically sensitive landscapes that are frequently subjected to agricultural activities, urbanization, and environmental challenges such as flooding and salinity. Accurate soil information in these areas is vital for optimizing land use, enhancing crop productivity, managing water resources efficiently, and mitigating environmental impacts.

To the best of the authors' knowledge, there has not been any literature review on DSM activities in lowland areas. Therefore, this article provides a comprehensive review of various advances in DSM approaches specifically for lowland areas. To comprehensively assess and synthesize the existing body of literature regarding the application of DSM approaches for soil mapping in lowland areas, we followed a systematic mapping approach as explained by James et al. [13]. It is intended that this review of the relevant literature will assist prospective researchers by identifying knowledge gaps in DSM approaches in lowlands, thereby guiding the path toward more robust and reliable soil information for improved land management, agricultural productivity, and environmental conservation. As the nexus of technology and soil science continues to evolve, embracing the potential of DSM in lowland areas not only enhances our understanding of these ecologically sensitive landscapes but also empowers policymakers, land managers, and researchers with the tools needed to make informed decisions for a sustainable future.

## 2. Soils in Lowland Areas

Soils in lowland or low-relief areas refer to specific types of soil found in low-lying regions, such as plains, river valleys, former flat glacial, floodplains, coastal plains, and alluvial valleys [14]. These areas are typically characterized by flat topography and relatively shallow topsoil with high bulk density [15]. They are mostly located between higher elevation regions and bodies of water, making them essential for agriculture, settlement, and various environmental functions. Soils in lowland areas exhibit distinctive characteristics that are important for mapping purposes. The key aspects to consider are as follows.

i.   Soil Hydrology: Lowland areas tend to have unique drainage patterns because of their relatively flat topography and proximity to water bodies, such as rivers, lakes, or coastal regions. Consequently, soils in lowland areas often exhibit distinct hydrological properties such as low internal drainage and a higher potential for waterlogging [16]. Understanding these characteristics is crucial for mapping purposes, as they help identify areas prone to flooding, soil moisture variations, and the overall drainage capacity of the soil.

ii. Organic Matter Accumulation: Lowland areas often experience high rates of organic matter accumulation, which often improve the soils' structure, mitigating the low drainage and limited oxygen availability. Waterlogging and limited oxygen may instead be given by the presence of fine textural soils and/or by the presence of depressional landforms typical of lowlands, and/or by the presence of shallow water tables [17]. As a result, these soils have unique properties and fertility profiles. The proper mapping of the organic matter content in lowland areas is vital for understanding nutrient cycling, carbon sequestration potential, and sustainable land management practices.

iii. Sediment Deposition: Lowland areas often serve as deposition sites for sediments carried by wind and water bodies, such as rivers, during flooding events [17]. These sediment deposits can lead to variations in the soil composition, specific properties, and nutrients across the landscape [18]. Mapping these variations helps to characterize soil formation processes, identify suitable land use practices, and manage erosion risks in lowland areas.

iv. Peat Soils: Peat soils may be prevalent in certain lowland areas [19]. These soils were formed through the accumulation of partially decomposed organic matter. Peat soils have specific properties such as high water-holding capacity, low bulk density, and acidic pH. Mapping peat soil distribution in lowland areas is crucial for understanding carbon storage, wetland conservation, and sustainable land-use planning.

v. Soil Salinity and Alkalinity: Some lowland areas, especially those in coastal regions or near saltwater bodies, may contain soils with elevated salinity or alkalinity levels [20]. These conditions can affect the growth and productivity of the vegetation and agricultural crops. Mapping the extent of soil salinity and alkalinity in lowland areas provides valuable information for site-specific soil management, irrigation practices, and land suitability assessment.

## 3. Materials and Methods

The systematic approach discussed by James et al. [13] was followed to compile the relevant information from the existing published papers with the aid of HubMeta software (https://hubmeta.com/, accessed date 1 August 2023) [21]. This approach involves a comprehensive process including team establishment, defining scope and questions, setting inclusion criteria, evidence search, screening, database creation, optional critical appraisal, findings description and visualization, and report production. In this study, a systematic search was conducted across two databases, Web of Science (WoS) and Scopus® (Figure 1). The aim was to identify fully published peer-reviewed journal articles in the English language that focus on the digital mapping of soil properties/classes in lowland areas. The two databases were queried using various search expressions built using standard Boolean operators. The search was without timespan restriction and, hence, covered publications from the period from 1991 to June 2023. Search strings were selected in such a way that most papers of our interest would be included. All search expressions were chosen based on the following defined keywords query for 'title' and 'keywords' (TOPIC): ("digital soil mapping" OR "soil mapping" OR "spatial distribution") AND ("lowland" OR "low-relief" OR "plain") AND "soil map".

The resulting papers were screened based on the criteria for the inclusion of DSM studies conducted in lowland or low-relief or plain areas. The exclusion criteria were as follows: (1) duplicates, (2) articles which did not predict soil properties or classes specifically in lowland areas, and (3) articles which adopted only geostatistical methods of DSM without considering any environmental covariates (SCORPAN). After applying the inclusion and exclusion criteria, only the articles whose focus include the mapping of soil in lowland or plain or low relief areas were targeted for systematic review.

From the selected papers, relevant information from these articles including the country, year of publication, target variable, land use, number of soil samples, method of sampling, validation techniques, environmental covariates, sources of environmental

covariates, DSM predictive approach/model, assessment metric, and the objective of the paper were recorded and presented in tables and appropriate maps to show the knowledge gaps and clusters in this research area.

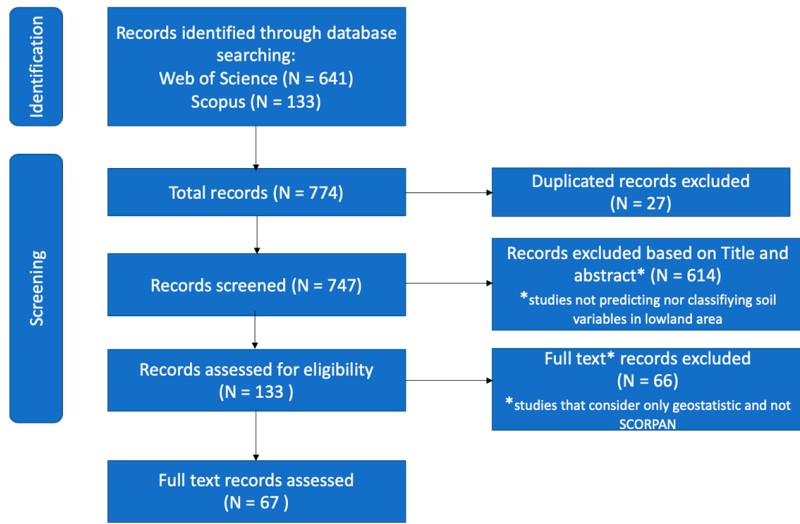

**Figure 1.** Schematic overview of the screening process applied to the articles examined for this study.

A total of 774 articles were found—641 in Web of Science and 133 in the Scopus database—using the search expressions (Figure 1). After the duplicate articles were removed, we investigated the remaining 747 articles to select the articles that met our relevant criteria. A total of 133 articles were selected after conducting title and abstract screening, and a total of 67 articles were found to meet all our criteria after completing a full text review of the articles. The collection of the evidence compiled, also known as the systematic map, is presented in a tabular format in Table 1 also in Supplementary: Table S1.

**Table 1.** Summary of remaining reviewed published papers on digital soil mapping in lowland/plains/low-relief areas.

| S/N | Reference | Target Soil Variables | Land Use | Environmental Covariate Combinations [Source] | DSM Models (Best Model in Comparison Studies Bolden) | Assessment Metric Combination | Validation Approach |
|---|---|---|---|---|---|---|---|
| | | | | *Traditional statistical approach* | | | |
| 1. | Yahiaoui et al. [22] | Soil salinity | Cropland | S [RS, EC], O [RS], R | Step MLR | | |
| 2. | Nawar et al. [23] | Soil salinity | Cropland | S [SS, RS] | PLSR, **MARS** | $R^2$ and RMSE | Independent validation |
| 3. | Cheng-Zhi et al. [24] | SOM | Cropland | R | **FSPW**, MLR | CCC, MAE, and RMSE | Independent Validation |
| 4. | [25] | Soil salinity variable (EC), clay content and SOM | Cropland | S [MRS] | PLSR, **MARS** | $R^2$, RMSE, and RPD | Data splitting |
| 5. | Vaudour et al. [26] | SOC, pH, CEC, Iron, Clay, Sand, Silt, CaCO$_3$ | Cropland | S [RS], O [RS] | PLSR | $R^2$, RMSE, and RPD | K-fold CV |
| 6. | Zhang et al. [27] | SOM | Cropland | S [RS], O [RS] | Step MLR | $R^2$, RMSE, and MAE | Data splitting |
| 7. | Buscaroli et al. [28] | Trace elements | Croplands, Urban and industrial areas | S [WDXRF] | PCA, CA | | |

<div align="center">**Table 1.** *Cont.*</div>

| S/N | Reference | Target Soil Variables | Land Use | Environmental Covariate Combinations [Source] | DSM Models (Best Model in Comparison Studies Bolden) | Assessment Metric Combination | Validation Approach |
|---|---|---|---|---|---|---|---|
| | | | | *Traditional statistical approach* | | | |
| 8. | Tang et al. [29] | SOM | Croplands | S, O [MRS] | **Step-MLR**, PLSR | $R^2$ and RMSE | Data splitting |
| 9. | Yu et al. [30] | Soil salinity | Croplands, grasslands, woodland | S [RS], O [RS, LU], R | PLSR | $R^2$, Bias, RMSE | K-fold CV |
| 10. | Ma et al. [31] | SOM | Croplands, Paddy field, forest | O [RS] | PLSR | $R^2$ and RMSE | LOOCV |
| | | | | *Geospatial and multivariate geostatistics approach* | | | |
| 11. | Lagacherie et al. [32] | Clay | Vineyard | S [HRS] | Co-kriging, block co-kriging | RMSE | K-fold CV |
| 12. | Bilgili [33] | Soil salinity variables | Croplands | R | OK, **RK**, KED, DK | RMSE, RI, Kappa | Data splitting |
| 13. | Zhao et al. [34] | SOM | Paddy field | O [RS] | OK, **RK** | RMSE, MAE, ME | LOOCV |
| 14. | Liu et al. [35] | SOC | Cropland | C, O, R | OK, SLR | MAE, RMSE, $R^2$ | Data splitting |
| 15. | Shabou et al. [36] | Soil texture class, Clay | Cropland, fruit trees | S [LS], O [MTD] | **Cokriging** | RMSE, $R^2$ | Independent validation |
| 16. | Walker et al. [37] | Clay, $CaCO_2$, EC, Iron, Sand, Silt, pH | Vineyard | S [LS], O [HS] | OK, CoKriging with CED | $R^2$ | LOOCV |
| | | | | *Statistical machine learning approach* | | | |
| 17. | Barthold et al. [38] | Soil nutrient: K and Mg | Forest | O, R, P | CART | - | K-fold CV |
| 18. | Mosleh et al. [39] | Sand, silt, clay, EC, CFs, SOC, pH and $CaCO_3$ | Cropland | S [LS], C, O [RS], R, P, A | ANN, BRT, MLR, **GLM** | RMSE, ME, $R^2$ | Data splitting |
| 19. | Mosleh et al. [40] | Soil taxonomy classes | Cropland | S [LS], C, O [RS], R, P, A | **RF**, MLR, ANN, BRT | Kappa, OA, Adjusted Kappa, Brier score | Data splitting |
| 20. | Pahlavan-Rad et al. [41] | SOC | Cropland | S, O [RS, LU], R | RF | RMSE, and MAE | K-fold CV |
| 21. | Pahlavan-Rad and Akbari-moghaddam [42] | Sand, silt, clay, pH | Cropland | O [RS], R | RF | RMSE, MAE, and ME | Data splitting, Independent validation |
| 22. | Mirakzehi et al. [43] | Soil taxonomy classes | Cropland | S [RS], R, O [RS] | RF | Kappa, OA | Data splitting, K-fold CV |
| 23. | Jamshidi et al. [44] | Soil taxonomy classes | Cropland, forest, grassland | O [LU, RS], R, P | DSMART | OA, CI | Independent validation |
| 24. | Zeng et al. [45] | Sand, Clay | | R [LSDF, RS] | RF | RMSE, MAE | LOOCV |
| 25. | Donoghue et al. [46] | pH, Clay, SOM, other soil nutrients | | | CA | | |
| 26. | Esfandiarpour-Boroujeni, Shamsabadi et al. [47] | Soil taxonomy class, soil WRB class | Cropland | S [LS, RS], R, P, A | **DT**, LVQ (ANN) | PPE | Data splitting |

**Table 1.** *Cont.*

| S/N | Reference | Target Soil Variables | Land Use | Environmental Covariate Combinations [Source] | DSM Models (Best Model in Comparison Studies Bolden) | Assessment Metric Combination | Validation Approach |
|---|---|---|---|---|---|---|---|
| | | | | *Statistical machine learning approach* | | | |
| 27. | Fathizad et al. [48] | SOC, EC, HM, AS | | S [RS], O [RS, LU], R, P | RF | MAE, RMSE, and $R^2$ | Data splitting |
| 28. | Esfandiarpour-Boroujeni, Shahini-Shamsabadi et al. [49] | Soil taxonomy class, soil WRB class | Cropland | S [LS, RS], R, P, A | ANN, **DT**, RF, SVM | OA, CI | Data splitting |
| 29. | Goldman et al. [50] | Soil texture class | Cropland, forest, Urban area | S [LS], R | RF | Kappa, OA, CI | Independent validation |
| 30. | Zare et al. [51] | ES, clay, sand, CEC | Cropland | S | SVM | CCC | LOOCV |
| 31. | Parsaie et al. [52] | Sand, Silt, Clay, $CaCO_3$, SOC | Cropland, rangeland | O [RS], R | Cubist, **RF**, DT | RMSE, MSE, $R^2$ | Data splitting |
| 32. | Wang et al. [53] | SOC | Cropland | S [RS] | RF, ANN, SVM, PLSR | RMSE, RPD | Data splitting |
| 33. | Abedi et al. [54] | Soil salinity variables (EC, SAR) | Cropland, Orchards | S [RS], R | DT, kNN, SVM, **Cubist**, **RF**, XGBoost | RMSE, MAE, $R^2$ | K-fold CV |
| 34. | Nabiollahi et al. [20] | pH, Soil salinity variables (EC, SAR) | Croplands | S [RS], O [LU, RS], R, P, A | RF | CCC, MAE, RMSE | K-fold CV |
| 35. | Habibi et al. [55] | Soil salinity variables (EC) | | S [RS], O [RS], R | ANN | MSE, $R^2$ | Data splitting |
| 36. | Rainford et al. [56] | SOC | Cropland, rangeland, forest, Urban area | C, O [LU], R, P, A | RF | RMSE, ME | Data splitting |
| 37. | Zhang et al. [57] | SOM | Cropland | S [RS], O [RS], R | **RF**, ANN, SVM | ME, RMSE, $R^2$ | Data splitting |
| 38. | Sothe et al. [58] | SOC | Forest | S, C, R, O [RS, SAR] | RF | RMSE, MAE, $R^2$ | Data splitting |
| 39. | Fathizad et al. [59] | SOC | Cropland | O [RS] | **RF**, SVM, ANN | RMSE, MAE, $R^2$ | K-fold CV |
| 40. | Zhang et al. [60] | SOC | Cropland | S [RS], C, O [RS], R, P | Cubist, XGBoost, **RF** | RMSE, $R^2$ | Independent validation |
| 41. | Luo et al. [61] | SOM | Cropland | O [RS, MTD] | RF | RMSE, $R^2$ | Data splitting |
| 42. | Zeng et al. [62] | SOM | Cropland | C, O [RS], R | RF, **DL [LSM-ResNet]** | CCC, MAE, ME, RMSE, $R^2$ | Data splitting |
| 43. | Sorenson et al. [63] | Soil type class | Forest | S [RS, SAR], O [RS], R | RF | Kappa | Independent validation |
| 44. | Xu et al. [64] | SOC | Cropland | S [RS], O [RS, MTD] | **RF**, Cubist, GBM | Bias, RMSE, $R^2$ | Data splitting |
| 45. | Haq et al. [65] | Soil texture class | Cropland | O [RS] | **RF**, SVM, LMT | OA, F1 score | K-fold CV |
| 46. | Wang et al. [66] | SOM | Paddy field | S [VNIR], O [VNIR, LU] | RF | RMSE, $R^2$ | Data splitting |
| 47. | Ge et al. [67] | Soil salinity variables | Cropland | S [RS], O [RS] | **Cubist**, RF, SVM, XGBoost | RMSE, $R^2$, MAE | Data splitting |

**Table 1.** *Cont.*

| S/N | Reference | Target Soil Variables | Land Use | Environmental Covariate Combinations [Source] | DSM Models (Best Model in Comparison Studies Bolden) | Assessment Metric Combination | Validation Approach |
|---|---|---|---|---|---|---|---|
| | | | | *Statistical machine learning approach* | | | |
| 48. | Lotfollahi et al. [68] | $CaCO_3$ | Cropland, rangeland | O [RS], R | **RF**, DT | RMSE, $R^2$ | Data splitting |
| 49. | Liu et al. [69] | SOC | Cropland | C, R | **RF**, SVM | Bias, RMSE, $R^2$ | K-fold CV |
| 50. | Adeniyi et al. [70] | Sand, Silt, Clay, pH, SOC, topsoil depth | Cropland, paddy field | O [LU], R | Cubist, GBM, GLM, **RF**, SVM, EL | CCC, RMSE | nestedCV |
| 51. | Dasgupta et al. [71] | Soil micronutrients | Cropland | S [RS], C, O [RS], R | EL, SVM, Cubist, RF, QRF, rpart, Rpart2, XGBoost, extraTrees, XCG, glmStepAIC, C LASSO, MARS | CCC, RMSE, MAPE | Data splitting |
| | | | | *Hybrid model approach* | | | |
| 52. | Mousavi et al. [72] | $CaCO_3$, Silt, Clay, pH, SOC, Sand | Cropland | R, O [RS] | **RF-RK** | Bias, CCC, RMSE, $R^2$ | Data splitting |
| 53. | Kumar et al. [73] | SOC | Forest | O [RS], R | RK (MLR-OK) | RMSE, ME | Data splitting |
| | | | | *Multi-approach methods* | | | |
| 54. | Maino et al. [74] | Soil texture (Sand, Silt and Clay) | Cropland | S, P [Radiometric Data] | Step-MLR, **NLML** | $R^2$ | Data splitting |
| 55. | Lamichhane et al. [75] | SOC | Cropland | S [LS], C, O [LU, RS], R, P, A, N | RK, **RF** | CCC, ME, RMSE, $R^2$ | Data splitting |
| 56. | Zhang et al. [76] | SOC | Cropland, forest | O [RS] | Step-MLR, PLSR, **ANN**, OK, SVM | RMSE, $R^2$ | Data splitting |
| 57. | Guo et al. [77] | SOC, SBD | Cropland | O [HRS, RS] | **ELM**, PLSR | RPIQ, RMSE, $R^2$ | Data splitting |
| 58. | Kaya et al. [78] | SOC, Soil nutrient (P) | Cropland, Orchards | S, C, O [RS], R, P | Cubist, RF, RF-RK, **Cubist-RK** | NRMSE, RMSE, MAPE, CCC | Data splitting |
| 59. | Kaya et al. [79] | Soil salinity variable [EC] | Cropland | O [RS, LU], R, P | RF, SVM, **RF-RK**, SVM-RK | NRMSE, RMSE, CCC | Data splitting |
| 60. | Rahmani et al. [80] | SOM, CEC | Cropland | R | UK, Cubist, RF | ME, CCC, RMSE, $R^2$ | Data splitting |
| 61. | Wu et al. [81] | SOC | Cropland, Paddy field, grassland, woodland | S, C, O [LU, RS], R | **Cubist**, OK, RF, Step-MLR | MAE, CCC, RMSE, $R^2$ | Data splitting |
| 62. | Yan et al. [82] | SOM | Cropland | S [HRS] | OK, **RF** | RPD, RMSE, $R^2$ | Independent validation |
| 63. | Chagas et al. [83] | Sand, silt, Clay | | O [RS] | MLR, **RF** | RMSE, $R^2$ | Data splitting |
| 64. | Samarkhanov et al. [84] | Soil salinity variable [EC] | Cropland | S [RS], O [RS] | **KNN**, MLR, PLSR | RMSE, $R^2$ | Data splitting |
| 65. | Shahrayini & Noroozi, [85] | Soil salinity variable [EC, SAR] | Cropland, rangeland | R, O [RS] | Step-MLR, **RF** | RMSE, $R^2$ | Data splitting |

**Table 1.** *Cont.*

| S/N | Reference | Target Soil Variables | Land Use | Environmental Covariate Combinations [Source] | DSM Models (Best Model in Comparison Studies Bolden) | Assessment Metric Combination | Validation Approach |
|---|---|---|---|---|---|---|---|
| | | | | *Multi-approach methods* | | | |
| 66. | Huang et al. [86] | EC, pH | Cropland, rangeland | R [PS] | Fuzzy k-means | RMSE, ME | |
| 67. | Huang et al. [87] | EC, pH | Cropland, rangeland | R, N | MLR, **REML**, OK | MSE | |

The best performing models in each study were bolden. Description of properties. Target soil variables: electrical conductivity (EC), sodium absorption ratio (SAR), soil organic carbon (SOC), soil organic matter (SOM), phosphorus (P), soil bulk density (SBD), coastal acid sulphate soils (CASS), calcium carbonate ($CaCO_3$), cations and cation exchange capacity (CEC), total nitrogen (TN), coarse fragments (CF), heavy metals (HM). Environmental covariates: soil (S), climate (C), organisms (O), relief (R), and parent material (P), age (A), and easting and northing coordinates/position (N). Sources: legacy soil map (LS), land use (LU), land surface dynamic feedback (LSDF), hyperspectral remote sensing data (HRS), multispectral remote sensing (MRS), near-infrared spectroscopy (NIR), remote sensing (RS), synthetic aperture radar (SAR), visible/near-infrared spectroscopy (VNIR), wavelength dispersive X-ray fluorescence (WDXRF), moderate resolution imaging spectroradiometer (MODIS) Terra MOD09A1. Evaluation metrics: coefficient of determination (R2), concordance correlation coefficient (CCC), mean absolute error (MAE), root mean squared error (RMSE), overall accuracy (OA), and Kappa index. DSM models: artificial neural network (ANN), boosted regression trees (BRT), clustering analysis (CA), classification and regression trees (CART), decision trees (DT), deep learning (DL), disaggregation and harmonisation of soil map units through resampled classification trees (DSMART), extreme learning machine (ELM), ensemble learning (EL), extremely randomized trees (extraTrees) least absolute shrinkage and selection separator (LASSO), linear regression with stepwise selection (leapSeq), K-nearest neighbours (KNN), partial least squares regression (PLSR), multivariate adaptive regression splines (MARS), multiple linear regression (MLR), principal component analysis (PCA), recursive partitioning and regression trees (rpart), support vector machines (SVM), OK, LSM-ResNet, residual maximum likelihood (REML), quantile regression forest (QRF), random forest (RF), extreme gradient boosting (XGBoost), and Gblinear booster (XGB).

## 4. Results and Discussion

### 4.1. Emergence of Interest and Growing Importance

Figure 2 exhibits the trend of the number of articles that focused on DSM in lowland areas. The distribution of selected articles according to the year of publication showed a consistent upward trend from 2013 to 2022, with the highest number of 16 articles in 2022 and 11 articles published by mid-year 2023.

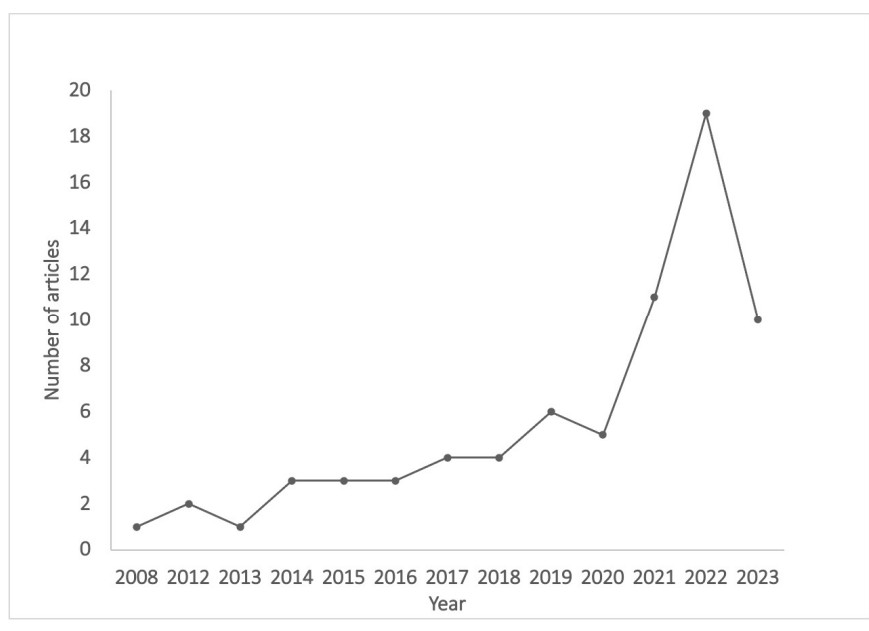

**Figure 2.** Trend of the number of articles published.

The temporal trend analysis of the selected articles demonstrated a growing interest in the application of soil mapping approaches in lowland areas over the past two decades. It also indicates the growing recognition of the need for accurate soil characterization in these environments. Lowland areas are characterized by ecological sensitivity and challenges related to flooding, salinity, and agricultural productivity. The rising interest in DSM underscores the importance of understanding soil properties and their spatial variations in addressing these multifaceted challenges. Moreover, the recent availability of high-resolution satellite data has contributed to the surge in DSM studies in lowland areas. For example, until 2014, the global coverage of the SRTM DEM was at a 90 m resolution, but since then, a 30 m version of the same elevation model was released worldwide.

Figure 3 displays the geographical distribution of the number of articles published over the period of this study. Out of 67 articles, the study areas of 22 articles were in China, followed by 18 in Iran, and 5 in the USA. Smaller proportions of articles were distributed across France, India, Italy, Canada, Brazil, Egypt, Turkey, Algeria, and Tunisia, indicating a global interest in lowland DSM.

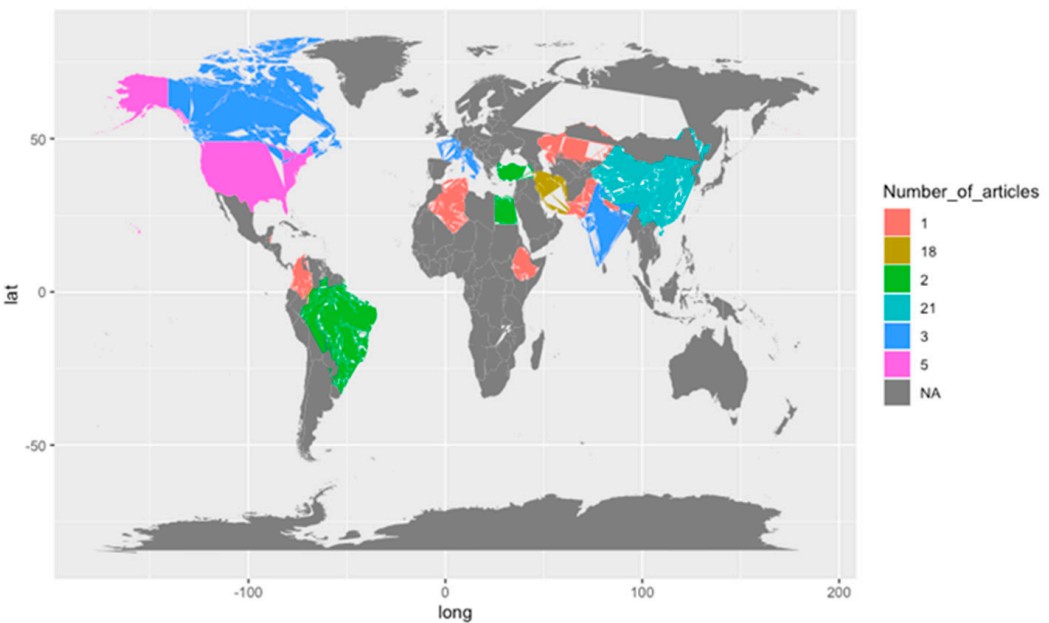

**Figure 3.** Geographic distribution of the number of articles published.

### 4.2. Dominant Land Use Categories

Figure 4 shows the common land use of the study areas where the DSM approach has been used for soil mapping in lowland areas from the published articles. Land use distribution within the selected articles demonstrated a varied focus. Agricultural cropland constituted the highest proportion, appearing in 62% of the total articles. The emphasis on DSM within cropland areas signifies the recognition of the intimate relationship between soil attributes and agricultural productivity. Accurate soil mapping in croplands aids in optimizing irrigation, fertilizer application, and crop selection, thereby contributing to efficient resource utilization and yield enhancement. In addition, the focus on woodland/tree areas (14% of the total articles) reflects the interest in understanding soil dynamics within these ecologically sensitive areas. DSM within woodland lowland areas helps in assessing soil erosion risks, determining soil nutrient availability for plant growth, and guiding forest management practices. This knowledge is vital for maintaining the ecological integrity of forest ecosystems and promoting sustainable forestry practices.

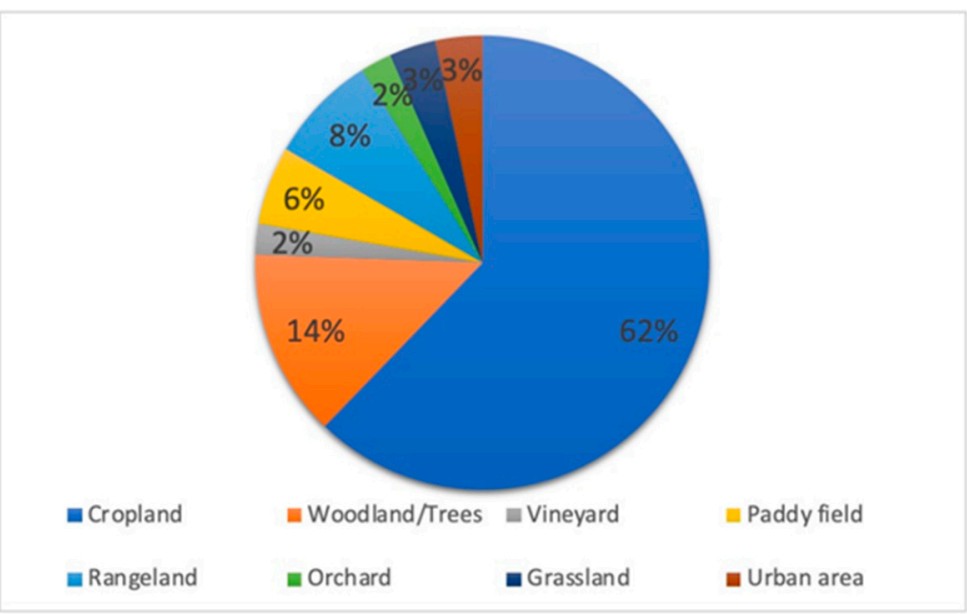

**Figure 4.** Percentage of land use from the articles published.

*4.3. Targeted Soil Variables in Lowland Areas*

Figure 5 represent frequency of predicted variables in different DSM articles in lowland areas. A total of 46% of the articles focused on predicting a single target soil variable and the corresponding digital soil map. This approach may reflect a pragmatic strategy to address specific soil-related challenges, e.g., SOC stock, soil salinity, etc. On the other hand, 38 out of the 67 articles (54%) aimed to predict multiple target soil variables and generate comprehensive digital soil maps. This emphasis on multifaceted soil variables signifies the increasing recognition of the interconnectivity between different soil attributes and the importance of capturing this complexity in mapping efforts. A total of 21% of the articles focused on mapping SOC-related properties such as SOC density, SOC stock, etc. Among the studied articles, SOC stood out as the most extensively studied variable. This prominence likely stems from the crucial role of SOC in determining soil fertility, carbon sequestration potential, and overall soil health [88,89]. Additionally, SOC content as well as SOM (which was 13% of the articles) can be a key indicator of land use sustainability and climate change mitigation strategies [90]. Similarly, the attention given to the mapping of sand, silt, and clay contents (14% of the articles) reflects the significance of soil texture in determining soil structure, water-holding capacity, and nutrient retention. Notably, soil salinity variables (15% of the articles) such as EC and SAR are also addressed, indicating the importance of understanding soil salt concentrations in lowland areas, where salinity can significantly impact plant growth, land use and land degradation [91–93]. Nutrient mapping, encompassing both macro and micronutrients, constitutes only 6% of the studies. Given the critical role of nutrients in agricultural productivity and ecosystem functioning, this presents an avenue for future research to investigate nutrient dynamics in lowland soils. Similarly, the limited attention (8%) directed towards soil class mapping, encompassing soil texture and taxonomy classifications, underscores an opportunity to more deeply explore the characterization of soil types within lowland environments. Accurate soil class mapping aids in informed decision-making related to agriculture, environmental conservation, and urban development.

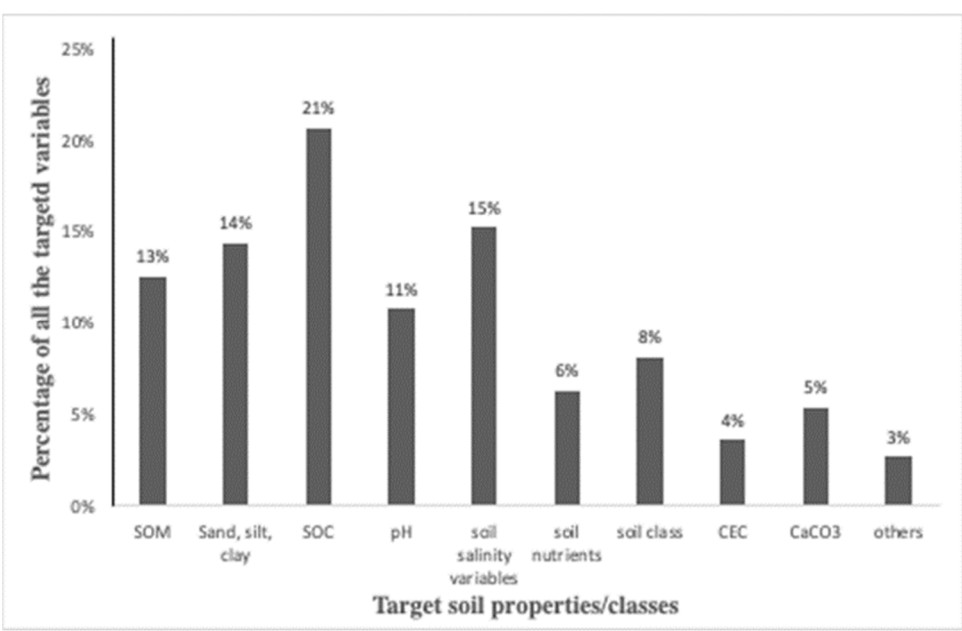

**Figure 5.** Percentage of targeted variables in the articles reviewed.

*4.4. Environmental Covariates for DSM in Lowland Areas*

Relevant environmental covariates can improve the accuracy of DSM [7]. The legacy soil maps, climatic data, digital elevation models (DEM), geology maps, remote sensing products, land use map, and geomorphological maps have been used as sources of environmental covariates (SCORPAN factors) in DSM activities in lowland areas are presented in Table 1. Figure 6 shows the frequency of the SCORPAN factors as covariates to predict a soil property or class in all the selected articles.

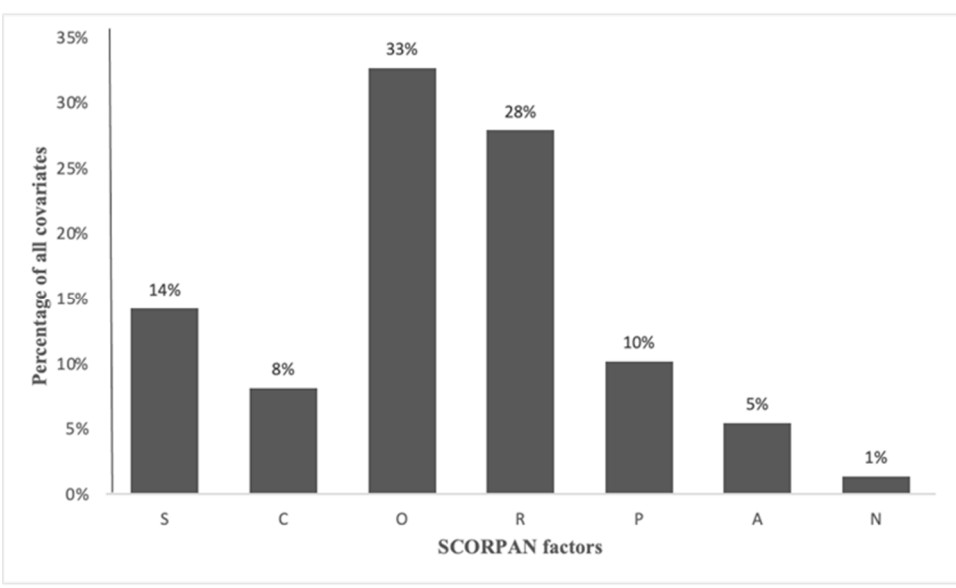

**Figure 6.** Percentage of environmental covariates in the articles reviewed.

Among the studied articles, the organism-related covariates (O) stood out to be the most extensively used (33% of the articles). This underscores the role of vegetation in shaping soil characteristics. Lowland areas are mostly agricultural areas. Agricultural practices such as tillage and other human interference weaken the relationship between vegetation and soil conditions [34,94]. Mapping soils in lowland areas presents specific challenges, owing to the unique characteristics of these landscapes. However, this review

study shows that vegetative spectral indices such as normalized differential vegetative index, enhanced vegetative index, soil adjusted vegetative index, etc., derived from remote sensing imagery such as Landsat 8 or Sentinel 2, are powerful covariates in mapping soils in lowland areas. Vegetative spectral indices and reflectance band data provide insights into land cover and vegetation health. Also, land use maps were considered as a good source of human interference information in lowland areas [70]. In farm-scale mapping, existing land use practices emerge as a significant governing element [95]. These covariates are valuable for understanding how plant communities impact soil properties through factors like root structure, nutrient cycling, and organic matter input in lowland areas.

The integration of relief-related covariates (R) demonstrates the importance of topography in influencing soil distribution and properties. It was used in 28% of the articles. Terrain attributes derived from digital elevation models (DEM) offer terrain information. The terrain attributes include elevation, the multi-resolution index of valley bottom flatness, the multi-resolution index of ridge top flatness, wetness index, mass balance index, the slope length and steepness factor of universal soil loss equation, mid-slope position, terrain ruggedness index, valley depth, vertical distance to channel network, etc. These indices are crucial for understanding soil erosion potential, water drainage patterns, and the accumulation of organic material in different landscape positions [96]. Cheng-Zhi et al. [24] proposed a technique for calculating fuzzy slope positions by assessing their similarity to standard slope positions. They employed this method in the digital mapping of soil organic carbon (SOC) content. Their research demonstrated improved mapping accuracy using the fuzzy slope position variable, coupled with a restricted set of soil samples, when compared to the utilization of conventional terrain parameters along with additional soil samples.

The soil-related covariates (S) (14% of the articles) indicate a strong interest in utilizing soil spectral information from remote sensing as well as proximal sensing techniques like soil spectrometers and existing soil maps (legacy soil maps) in lowland areas. Soil spectral indices and reflectance data enable researchers to capture the unique spectral signatures of soil characteristics. Soil spectral indices include, among others, bare soil index, brightness index, normalized difference soil index, etc. This approach is particularly effective for estimating soil attributes like organic matter content, mineral composition, and soil salinity variables. Some of the commonly extracted environmental covariates from legacy soil maps include soil type, group, texture, landform, drainage, and physiography. However, it is essential to consider the spatial scale and cartographic scope of the existing soil maps before employing them for DSM [97].

Furthermore, climate-related covariates which focus on climatic factors were found in 8% of the reviewed articles. These covariates were recognized for their significance in shaping soil properties, especially in lowlands with diverse climatic conditions [35,49,62]. They play a critical role in assessing soil resilience to climate change and its implications for sustainable land use and agriculture. Parent-related covariates constituted 10% of the articles and encompassed factors related to soil's geological and pedological history, including parent material composition. Their limited use might be due to the perception that lowland areas often have uniform parent material, although exceptions exist in regions with complex geological histories. Age-related covariates, accounting for 5% of the articles, include factors related to soil development and age. While their usage was relatively limited, they offer valuable insights into soil dynamics, particularly in lowlands with dynamic histories of sediment deposition and landscape evolution. Lastly, position-related covariates (N), present in 1% of the articles, represent the spatial positioning of soil sampling points within lowland landscapes. Despite their infrequent use, these covariates provide essential information even in apparently uniform lowland environments, as microtopographic variations can impact soil attributes when combined with other landscape factors [75,87]. The study highlights the importance of tailoring covariate selection to specific research objectives and the complexities of the lowland landscapes, emphasizing their role in enhancing the accuracy of DSM in these critical regions.

Across various studies, the importance of variables in the DSM of lowland areas varies, reflecting the diversity of landscapes and the specific focus of each study. Terrain attributes such as channel network base level, valley depth, vertical distance to channel network, etc. consistently emerge as influential factors (Figure 7). For instance, in studies by Mosleh et al. [39,40] and Jamshidi et al. [44] terrain attributes were highlighted as the main predictors for soil properties and classes. Distance from rivers, often associated with topographic features, appeared critical in studies by Pahlavan-Rad et al. [41,42] and Mirakzehi et al. [43]. Additionally, spectral indices derived from remote sensing data (RS), such as NDVI, SAVI, and band information, frequently featured prominently, as seen in studies by Kumar et al. [73], Abedi et al. [54] and Parsaie et al. [52]. The results underline the significance of both terrain attributes and remote sensing data in understanding soil variability in lowland areas. To enhance DSM accuracies, incorporating a combination of terrain attributes and remote sensing data proves beneficial. Combining the strengths of both types of variables can provide a comprehensive understanding of soil distribution in lowland areas.

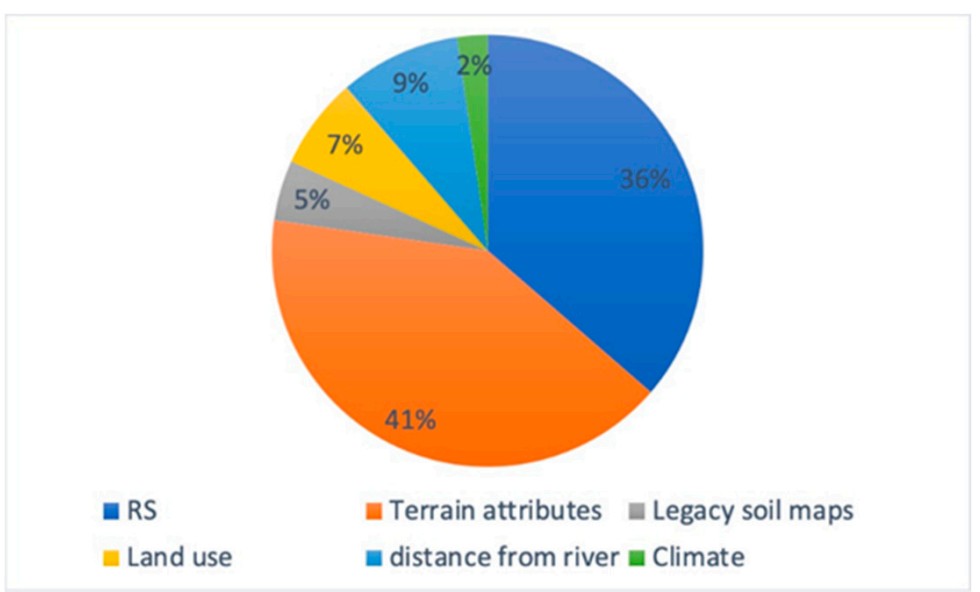

**Figure 7.** Percentage of important variables in the articles reviewed.

*4.5. DSM Approaches in Lowland Areas*

The successful implementation of DSM approaches in lowland areas requires a judicious selection of methodologies that account for the unique characteristics of these landscapes. Leveraging the power of technology and data science, modern DSM techniques offer the potential to overcome traditional limitations, enhance accuracy, and enable a broader spatial coverage. The approaches commonly employed in DSM can be generalized as belonging to four broad categories: (1) traditional statistical approaches [96] such as MLR, PLSR, etc.; (2) geospatial and multivariate geostatistical approaches such as cokriging, block kriging, OK, etc.; (3) statistical machine learning (ML) approaches such as RF, SVM, Cubist (Cu), DT, DL, etc.; and (4) hybrid model approaches such as RK, RFRK, etc. [95,98]. A total of 50% of the articles use only statistical ML approaches in their studies, 14% use traditional statical approaches, 13% use geospatial and multivariate geostatistical approaches, 3% use hybrid approaches, and 20% of the articles use all the approaches for their comparison studies.

Figure 8 displays the variety of DSM techniques utilized in the articles. RF was the most frequently used model, with 37 articles in the context of DSM in lowland areas. This was followed by cubist and DT models at 16. The diversity of predictive models used

underscores the complexity of soil systems and the importance of selecting appropriate models for accurate predictions.

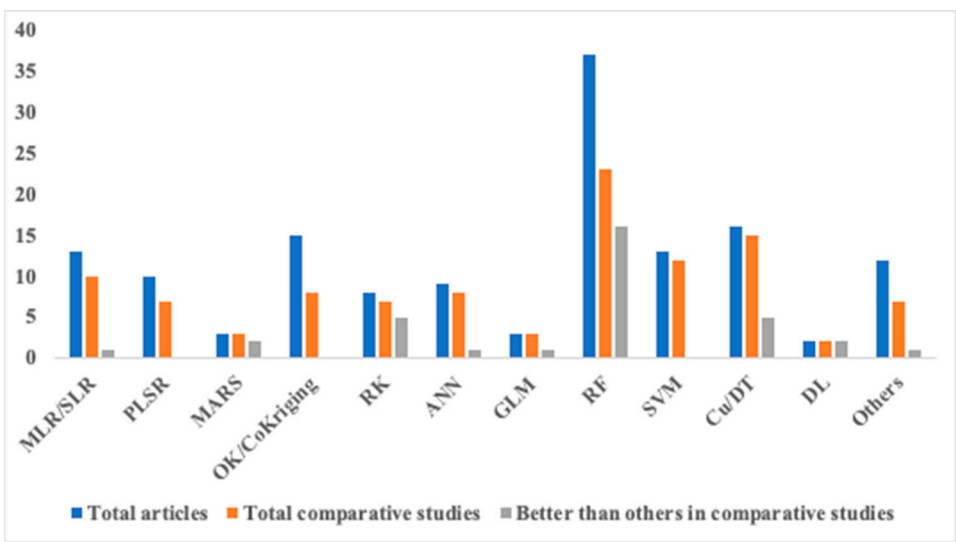

**Figure 8.** DSM models used in the reviewed articles.

Figure 8 also incorporates the number of articles that assessed different predictive algorithms, alongside the number of articles in which these algorithms demonstrated superior performance compared to others. This evaluation was grounded in the RMSE and error metrics, as indicated by the articles, employing data partitioning, cross-validation, and independent validation techniques. In most of the multi-approach comparative studies, statistical machine learning approaches often outperform other methods. However, in refs. [72,78,79], hybrid techniques which incorporate kriging of ML model residuals [99,100] were found to outperform ordinary ML models. The emerging role of hybrid models that combine geostatistical and ML approaches leverage the strengths of both methodologies, enhancing accuracy and prediction performance. This emphasizes the potential of hybrid models to capture spatial autocorrelation while benefiting from the predictive power of machine learning [101].

The diversity of DSM approaches employed, as depicted in Table 1 and Figure 5, is indicative of the multifaceted nature of soil systems and the recognition that no single model can effectively capture all variations. Altogether, RF emerges as the most frequently used model, indicating its adaptability and versatility in predicting soil properties across various landscapes. A total of 23 comparative studies compare the performance of RF with others. RF outperformed other predictive models in 16 of them. Cubist/DT models were the second most common models used in DSM in lowland areas, and 5 out of 15 comparative studies concluded that they are better than other models. MLR and SVM were used in at least 10 reviewed articles and other models outperformed them in all. Deep learning (DL) models are promising models that were used by 2 articles and performed comparatively better than another model in all. Other commonly used and promising models were RK, MARS which were used by at least 3 articles reviewed and performed comparatively better than other model in at least two studies.

The application of various algorithms, known as predictive models, is central to establishing quantitative relationships between input predictors (environmental covariates) and target soil variables. This process involves modelling a training dataset to regression and/or classification procedures [102]. In DSM, the utilization of high-level computer-based programming languages like R and Python has become prevalent for implementing diverse ML models. An increasingly prominent subset of ML algorithms in recent years is tree models [102]. Among these, CART serves as the basic form, constructing a tree-based structure of predictor variables for decision-making purposes. A more sophisticated iteration of

CART is the RF, which generates multiple decision trees from input variables instead of a single tree. The final decision results from an ensemble of these trees [102]. RF stands out for its capacity to handle sizable datasets, accommodate various data types, capture non-linear relationships, and process computations more swiftly [103]. The landscape of tree-based models is further enriched by options like BRT and cubist. Additionally, an extended form of the RF model, QRF, has found adoption in DSM studies in lowland areas (Table 1). ANN is another robust ML method for DSM in lowland areas. This technique involves three layers of neurons: input neurons (predictors), hidden neurons, and output neurons (target variable). ANN excels in establishing intricate non-linear relationships among covariates and handling complex datasets [103]. The progression of ANN techniques has given rise to deep learning (DL), an advanced iteration of neural networks, increasingly applied in recent DSM efforts in lowland areas. Additionally, ensemble methods have gained traction, involving the amalgamation of predictions from multiple ML models to produce a more accurate singular prediction. This ensemble approach has been growing in prominence in DSM applications in lowland areas [54,70,71].

*4.6. Evaluation of DSM Approaches*

Figure 9 displays evaluation (validation) techniques used in assessing the level of the map accuracy. This review identifies that 58% of DSM studies in lowland areas adopted a data splitting technique for model evaluation. Cross-validation and independent validation methods were adopted in 28% and 14% of the articles, respectively. Ref. [95] outlined three distinct evaluation approaches: cross-validation, data splitting, and independent validation. The data splitting technique involves partitioning the input dataset into training and testing subsets. These subsets are then employed for model calibration and validation, respectively. Cross-validation (CV) encompasses omitting either one observed value (leave-one-out method) or a subset of values (K-fold CV method) or looping an inner and an outer subset of values (nested CV) [104]. The remaining data are utilized to train the model for predicting the omitted values, serving as an evaluation measure. Independent validation necessitates the collection of additional samples through independent sampling for dedicated evaluation. In each of these approaches, the congruence between predicted and observed values is measured using appropriate metrics to gauge prediction accuracy. Nevertheless, the data-splitting technique is categorized as an internal assessment method, except when samples are acquired through a probability sampling approach [105].

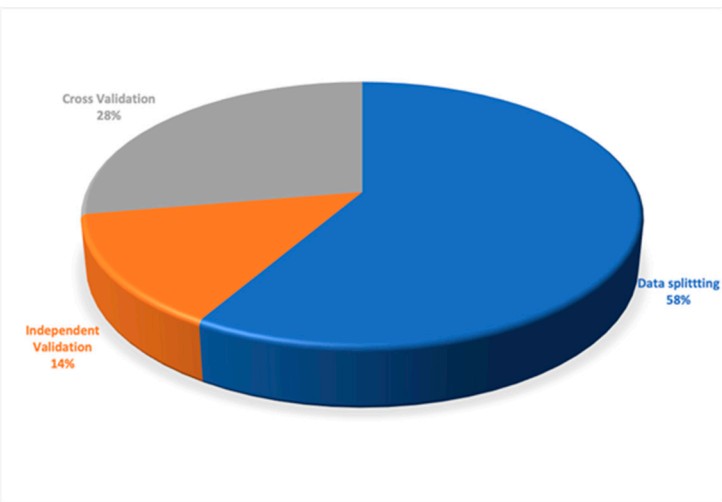

**Figure 9.** Evaluation techniques used in the reviewed articles.

Evaluation metrics like R2, CCC, MAE, and RMSE are commonly employed for soil continuous properties. These accuracy measures can fluctuate based on factors such as soil properties, depths, sample sizes, prediction models, and mapping approaches. Meanwhile,

metrics like OA and the Kappa index are commonly employed to evaluate soil classification such as soil taxonomy, soil texture, etc. Hence, effective strategies must be devised to enhance the precision of soil mapping predictions.

## 5. General Discussion and Outlook

In lowland areas, it might be tempting to assume that the soil properties remain uniform across the landscape. However, this assumption overlooks the fact that even in seemingly homogeneous terrains, there can be intricate variations in soil classes and properties, and these variations can manifest at various scales [106]. At a fine scale, which refers to relatively small and localized areas, variations can emerge due to a range of factors. For instance, micro-depressions in the landscape can collect and retain water differently than surrounding areas, leading to variations in soil moisture and properties [107]. Similarly, sediment deposition in particular spots, often associated with water bodies, can result in unique soil characteristics [108]. Hydrological processes, such as seasonal flooding or changes in groundwater levels, can also influence soil properties in specific locations [109–111]. These fine-scale variations, although they might appear minor in the broader context of lowland landscapes, are essential to consider when mapping and characterizing lowland soils accurately. Neglecting them could lead to oversimplified soil maps that fail to capture the subtleties of soil properties. Therefore, recognizing and accounting for these small-scale variations is essential for comprehensive and reliable DSM in lowland areas.

However, this systematic review has shed light on the evolution and current state of DSM in lowland areas. The growing interest in this field reflects the recognition of the crucial role that soil properties and classes play in lowland ecosystems and their impact on various land use practices. The number of identified articles (67 articles) suggests a relatively modest literature base, highlighting potential research gaps. Additionally, there are geographical biases, potentially limiting the generalisability of findings. Some land use categories remain underrepresented, indicating a need for more diverse studies. Also, the observed recent increase in publications on DSM in lowlands could be attributed to the latest advancements in producing high resolution DEMs. The vertical accuracy of DEMs, which provide crucial information for soil mapping, has only recently seen significant improvements with new products like LIDAR-based techniques or satellite-based information such as TerraSAR-X [112,113]. In lowlands, where elevation gradients are often quite small, these new DEM products provide a higher vertical resolution that can capture the subtle variations in elevation [114,115], which was a significant challenge in the past. With these finer-resolution DEMs, it was possible to represent the topography of lowland regions more accurately, leading to significantly improved soil mapping outcomes.

Furthermore, the existing literature on DSM in lowland areas reveals a significant knowledge gap concerning the nuanced role of specific environmental variables that could enhance mapping accuracy. While various studies highlight the importance of relief-related covariates derived from DEM (terrain attributes) and organism-related and soil information delineated from the spectral indices of remote sensing sensors, the precise identification and exploration of certain environmental covariates within these categories remain underexplored. The variability in lowland landscapes, influenced by factors such as micro-depressions, sediment deposition, and hydrological processes, suggests that there might be unique environmental variables contributing to soil heterogeneity. Understanding and incorporating these specific variables into DSM models is crucial for a more comprehensive and accurate mapping of soil properties in lowland areas, ultimately addressing the intricacies of these dynamic landscapes. Addressing these knowledge gaps holds the key to advancing the precision of DSM, facilitating improved land management, enhancing agricultural productivity, and contributing to effective environmental conservation strategies in lowland areas. Also, the adoption of various DSM approaches, especially random forest machine learning model and emerging deep learning techniques, reflects the

advancement of technology and data science in addressing soil variability challenges in recent decades.

The findings of this review suggest several avenues for future research. First, there is a need to further investigate the relationship between soil properties and land use practices, particularly in heterogeneous lowland landscapes. This is essential for sustainable agriculture, climate resilience, biodiversity conservation, and urban planning, ensuring a balance between human demands and environmental stewardship. Second, researchers should explore hybrid models that integrate geostatistical and machine learning techniques, including advanced approaches like deep learning, to enhance predictive accuracy in lowland ecosystems due to their inherent complexity. The complexity of spatial and temporal variations in these ecosystems can challenge traditional geostatistical models, but machine learning methods, capable of unveiling intricate patterns in both extensive and limited data, have the potential to enhance predictive accuracy [5,103], and support more informed ecological management choices in lowland areas. Additionally, further research is needed to comprehensively investigate how variations in data acquisition, model selection, and covariate choice may affect the accuracy and applicability of DSM, especially when transitioning from lowland areas to highlands or hilly areas with clear drainage patterns.

## 6. Conclusions

This systematic review focused on the dynamic landscape of digital soil mapping in lowland areas, shedding light on the current state, trends, and knowledge gaps within this field. Employing a comprehensive systematic approach, the study identified and analysed 67 relevant articles published between 2008 and June 2023. The emerging trend of increasing publications, particularly in recent years, underscores the growing recognition of the pivotal role DSM plays in understanding soil properties in lowland ecosystems. The identified knowledge gaps highlight the need for a nuanced exploration of specific environmental variables influencing soil heterogeneity in lowlands. While relief-related covariates, organism-related factors, and soil information from spectral indices have been recognized, the precise identification and exploration of unique environmental variables contributing to soil variability remain underexplored. The systematic map presented in Table 1 provides a structured compilation of key information from the selected articles, offering valuable insights into the distribution of studies across countries, land use categories, targeted soil variables, and employed DSM approaches. The observed dominance of agricultural cropland as the primary focus of DSM studies in lowlands reflects the intimate relationship between soil attributes and agricultural productivity. The significance of predicting multiple target soil variables, especially soil organic carbon, soil salinity, and soil texture, underscores the recognition of the interconnectedness of different soil attributes in lowland ecosystems. The extensive use of vegetation-related covariates emphasizes the pivotal role of vegetation in shaping soil characteristics in these areas. Furthermore, the incorporation of relief-related covariates, including terrain attributes derived from digital elevation models, highlights the importance of topography in influencing soil distribution and properties. The systematic evaluation of DSM approaches reveals the prevalence of statistical machine learning models, with random forest emerging as the most frequently used model, indicating its versatility in predicting soil properties across diverse lowland landscapes. This study emphasizes the significance of tailoring DSM approaches to the unique challenges posed by lowland areas, including limited soil samples, low topographic variability, and challenges associated with the scale and resolution of covariates. While data splitting is the most widely adopted technique, this study highlights the need for consistent evaluation metrics, considering variations in soil properties, depths, sample sizes, prediction models, and mapping approaches. Looking ahead, this systematic review suggests several avenues for future research. There is a pressing need to look deeper into the relationship between soil properties and land use practices, particularly in heterogeneous lowland landscapes. Exploring hybrid models that integrate geostatistical and machine learning techniques, including advanced approaches like deep learning, can

enhance predictive accuracy in the face of the inherent complexity of lowland ecosystems. Additionally, a more comprehensive investigation into the variations in data acquisition, model selection, and covariate choice is crucial for advancing the accuracy and applicability of DSM, especially during transitions from lowland to highland areas or areas with distinct drainage patterns. Addressing these research gaps holds the key to advancing the precision of DSM, facilitating improved land management, enhancing agricultural productivity, and contributing to effective environmental conservation strategies in lowland areas.

**Supplementary Materials:** The Supporting Information can be downloaded at: https://www.mdpi.com/article/10.3390/land13030379/s1. Table S1: Summary of reviewed published papers on digital soil mapping in lowland/plains/low-relief areas.

**Author Contributions:** Conceptualization, O.D.A. and M.M.; methodology, O.D.A. and H.B.; software, O.D.A. and H.B.; validation, O.D.A. and M.M.; formal analysis, O.D.A.; investigation, M.M.; resources, O.D.A.; data curation, O.D.A. and H.B.; writing—original draft preparation, O.D.A.; writing—review and editing, M.M. and O.D.A.; visualization, O.D.A.; supervision, M.M.; project administration, M.M.; funding acquisition, M.M. All authors have read and agreed to the published version of the manuscript.

**Funding:** This research was supported by Regione Lombardia, POR FESR 2014-2020—Call HUB Ricerca e Innovazione, Project 1139857 CE4WE: Approvvigionamento energetico e gestione della risorsa idrica nell'ottica dell'Economia Circolare (Circular Economy for Water and Energy) and the University of Pavia.

**Data Availability Statement:** The original contributions presented in the study are included in the article/supplementary material, further inquiries can be directed to the corresponding author.

**Acknowledgments:** I acknowledge Hauwa Bature for her involvement in the data retrieving and processing.

**Conflicts of Interest:** The authors declare no conflicts of interest.

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
