# Peer review of "A Systematic Review on Digital Soil Mapping Approaches in Lowland Areas"

_land, doi:10.3390/land13030379_

Round 1

Reviewer 1 Report

Comments and Suggestions for Authors

Dear Authors! I with interest read you manuscript (review) entitled: “A systematic review on Digital Soil Mapping Approaches in Lowland Areas”. The topic of review is relevant because digital soil mapping in plains and lowlands faces unique challenges, in particular: i) Plains and lowlands are often characterized by homogeneous soils, making it difficult to differentiate between them; ii) Lack of relief can lead to poor drainage and waterlogged soils, making mapping difficult; iii) High groundwater levels can affect soil characteristics and make it difficult to study. For example, the accuracy of mapping of soils in lowlands is 42% in the study (Comparative Assessment of Digital and Conventional Soil Mapping: A Case Study of the Southern Cis-Ural Region, Russia).

The presented article is fits to Land journal, also it suitable for consideration in other MDPI journals like ISPRS International Journal of Geo-Information, Soil Systems, Sustainability, etc. At almost the article is well written, however I have some comments and suggestions:

1.                L. 55. What about anthropogenic factor? which also in most cases is considered as one of soil forming factor.

2.                L. 110. Sediments also delivered from elevated areas.

3.                Table 1, could be presented in supplementary material. The information about soil type in this table will be interesting.

4.                The quality of Fig. 2 could be improved.

5.                For first time in using of abbreviations need to provide full names (SOM, EC, SAR, etc) in text or for Fig. 5. As well for Section 4.5 (Fig. 6).

6.                The reference list seems prepared not according to journal rules.

Author Response

Dear Reviewer,

Thank you for your valuable feedback on our manuscript. We sincerely appreciate the time and effort you dedicated to reviewing our work. Below are our responses to your comments:

1. We acknowledge the importance of the anthropogenic factor in soil formation, which is represented in our study by the "O" component in the SCORPAN formula. This component encompasses organisms, vegetation, and human activities, thereby providing a comprehensive analysis of soil formation factors.

2. While sediments can indeed be transported from elevated areas to lowlands, our study primarily focuses on the unique characteristics and challenges specific to lowland areas. However, we appreciate your point and will consider incorporating this aspect in future revisions.

3. Thank you for the recommendation regarding Table 1. We have additional details in the supplementary material, although it's worth noting that not all papers provided information on soil types.

4. We have taken note of your suggestion to improve the quality of Figure 2 for better visualization of the trend of published articles. We have made the necessary enhancements to ensure clarity and readability.

5. Regarding the expansion of abbreviations, we have provided the full names of the abbreviations under Table 1 to ensure clarity for readers. 

We appreciate your observation regarding the reference list. We have ensured that the references are formatted according to the guidelines provided by MDPI for consistency and adherence to journal rules.

Once again, we extend our gratitude for your insightful comments, which have undoubtedly contributed to improving the quality of our manuscript.

Best regards,
Odunayo David Adeniyi
Department of Earth and Environmental sciences, University of Pavia.

Reviewer 2 Report

Comments and Suggestions for Authors

The paper presents interesting results and will be useful for DSM community.

A lot of work was done. Below are my comments and suggestions.

Line 236. There are different tenses that you are using. Please fit as one tense when you describe results (past and present) throughout the article.

Lines 311-328. Please provide references for the mentioned papers. It is interesting to know what studies used climate, position, etc.

Line 362. The abbreviature was already mentioned.

373. RMSE is also an error metric.

388. A decision tree approach is a combination of decisions, and a random forest is a combination of many decision trees. Please correct as “cubist and other decision tree…”

439. The RMSE abbreviature was already mentioned before.

466-467. Where do you know about the limited availability of soil samples in lowland areas? You don’t specify what region (I agree that it exists for northern areas for Arctic and Antarctic regions, etc (see 10.1016/j.geodrs.2024.e00776)). Confirm this statement with references, or eliminated.

467-469. How flat area can pose challenges for soil sampling? It's strange because it works for mountain areas, etc.

469-472. If there is flat relief it does not mean that other SCORPAN factors do not work well. There are many studies that used just remote sensing products (see https://doi.org/10.3390/rs14122917). So, for flat regions, it is not a big problem!

466-476. The whole paragraph is fairly for general DSM, not only for lowland areas. So those are not “unique challenges” as you mentioned on line 466.

Author Response

Dear Reviewer,
Thank you for your thorough review of our manuscript. We appreciate your positive feedback and constructive comments, which will undoubtedly help enhance the quality of our work. Here are our responses to your specific points:

  1. We acknowledge the inconsistency in tenses and we will ensure to maintain a consistent tense, predominantly using the past tense when describing results.
  2. Regarding the mention of specific papers in lines 311-328, references has been provided to readers with further information on studies that utilized climate, position, etc., as suggested.
  3. Thank you for highlighting the redundancy in mentioning abbreviations and error metrics such as RMSE multiple times. We will rectify this oversight accordingly.
  4. You are correct in pointing out the distinction between decision tree approaches and random forests. We will revise the text to accurately reflect this. However, DT stands for single decision tree algorithm which is also use for regression and classification modelling.
  5. We have eliminate lines 466-476.

Thank you once again for your valuable feedback. We are committed to addressing these points to improve the overall quality and clarity of our manuscript.

Best regards,
Odunayo David Adeniyi
University of Pavia

Reviewer 3 Report

Comments and Suggestions for Authors

Digital soil mapping is important in soil science at large scale. This systematic review explores the DSM approaches in lowland areas by compiling and analysing 18 published articles from 2008 to mid-2023. Authors analyzed the trend of article published and revealed the targeted soil variables in lowland areas. It analyzed the environmental covariates and approaches of DSM in the lowland areas. The study concludes by outlining future research directions, highlighting the 27 urgency of understanding the intricacies of lowland soil mapping for improved land management, 28 heightened agricultural productivity, and effective environmental conservation strategies. Therefore, the idea is smart and it is well-structured with clear tables and figures. The writing is proper as well. 

Comments on the Quality of English Language

Digital soil mapping is important in soil science at large scale. This systematic review explores the DSM approaches in lowland areas by compiling and analysing 18 published articles from 2008 to mid-2023. Authors analyzed the trend of article published and revealed the targeted soil variables in lowland areas. It analyzed the environmental covariates and approaches of DSM in the lowland areas. The study concludes by outlining future research directions, highlighting the 27 urgency of understanding the intricacies of lowland soil mapping for improved land management, 28 heightened agricultural productivity, and effective environmental conservation strategies. Therefore, the idea is smart and it is well-structured with clear tables and figures. The writing is proper as well. 

Author Response

Dear Reviewer,
Thank you for your insightful comments on our manuscript. We are delighted to hear that you found the study's approach and structure to be well-conceived and executed, and that you appreciate the clarity of our presentation.

Your recognition of the importance of digital soil mapping (DSM) in soil science, particularly in lowland areas, aligns with our motivation for conducting this review. We aimed to provide a comprehensive analysis of DSM approaches in lowlands to contribute to improved land management, agricultural productivity, and environmental conservation efforts.

We are pleased that you found the presentation of trends in published articles, analysis of targeted soil variables, examination of environmental covariates, and discussion of future research directions to be valuable contributions to the field. 

Thank you for acknowledging the clarity of our tables and figures, as well as the overall quality of the writing. Your positive feedback is greatly appreciated, and we are grateful for your thoughtful review of our manuscript.

Best regards,
Odunayo David Adeniyi
University of Pavia

Round 2

Reviewer 2 Report

Comments and Suggestions for Authors

authors have corrected the issues